# Early-Onset Infantile Facioscapulohumeral Muscular Dystrophy: A Timely Review

**DOI:** 10.3390/ijms21207783

**Published:** 2020-10-21

**Authors:** Tai-Heng Chen, Yan-Zhang Wu, Yung-Hao Tseng

**Affiliations:** 1Section of Neurobiology, Department of Biological Sciences, University of Southern California, Los Angeles, CA 90089, USA; 2Department of Pediatrics, Division of Pediatric Emergency, Kaohsiung Medical University Hospital, Kaohsiung Medical University, Kaohsiung 80708, Taiwan; ijw168@yahoo.com.tw (Y.-Z.W.); G120042181@yahoo.com.tw (Y.-H.T.); 3School of Post-Baccalaureate Medicine, College of Medicine, Kaohsiung Medical University, Kaohsiung 80708, Taiwan

**Keywords:** facioscapulohumeral muscular dystrophy, early-onset, infantile FSHD, multidisciplinary care

## Abstract

Facioscapulohumeral muscular dystrophy (FSHD)—the worldwide third most common inherited muscular dystrophy caused by the heterozygous contraction of a 3.3 kb tandem repeat (D4Z4) on a chromosome with a 4q35 haplotype—is a progressive genetic myopathy with variable onset of symptoms, distribution of muscle weakness, and clinical severity. While much is known about the clinical course of adult FSHD, data on the early-onset infantile phenotype, especially on the progression of the disease, are relatively scarce. Contrary to the classical form, patients with infantile FSHD more often have a rapid decline in muscle wasting and systemic features with multiple extramuscular involvements. A rough correlation between the phenotypic severity of FSHD and the D4Z4 repeat size has been reported, and the majority of patients with infantile FSHD obtain a very short D4Z4 repeat length (one to three copies, *Eco*RI size 10–14 kb), in contrast to the classical, slowly progressive, form of FSHD (15–38 kb). With the increasing identifications of case reports and the advance in genetic diagnostics, recent studies have suggested that the infantile variant of FSHD is not a genetically separate entity but a part of the FSHD spectrum. Nevertheless, many questions about the clinical phenotype and natural history of infantile FSHD remain unanswered, limiting evidence-based clinical management. In this review, we summarize the updated research to gain insight into the clinical spectrum of infantile FSHD and raise views to improve recognition and understanding of its underlying pathomechanism, and further, to advance novel treatments and standard care methods.

## 1. Introduction

Facioscapulohumeral muscular dystrophy (FSHD; MIM158900) accounts for one in 15,000–20,000 normal individuals worldwide and is the third most common form of muscular dystrophy after Duchenne and myotonic muscular dystrophy [1]. In 1884, FSHD was first reported by Landouzy and Dejerine [2], who summarized a novel pattern of weakness affecting the facial (facio), shoulder (scapula), and upper arm (humeral) muscles. In 1952, a study of six generations of a family in Utah further described the clinical presentations of FSHD [3]. According to the diagnostic criteria proposed by the International Facioscapulohumeral Consortium [3,4], FSHD is generally defined by the characteristic, slow progression of initially restricted muscle weakness distribution (Table 1). However, because of the high degree of variability in clinical manifestations and the availability of advanced genetic diagnosis, molecular testing has almost replaced these clinical diagnostic criteria for FSHD [5].

## 2. Genetics and Pathogenesis of FSHD

Two genetically distinct types of FSHD, called FSHD type 1 (FSHD1) and type 2 (FSHD2), are classified. Despite having distinct genotypes, a dysregulated epigenetic repression of the retrogene, called double homeobox 4 (DUX4) in the skeletal muscle, plays a key role in the pathogenesis of both genotypes of FSHD [6]. The DUX4 protein is a transcription factor expressed in the germline, but its complete function is still under investigation. Several pieces of evidence show that DUX4 can regulate the expression of genes associated with stem cell and germline development [7]. Preliminary data suggest that inappropriate expression of DUX4 and its transcriptional targets in skeletal muscle can lead to apoptosis, impaired muscle regeneration (myogenesis), and induce inflammation [5,8]. Besides, DUX4 expression may be associated with clinical severity in FSHD. DUX4 expression, as well as DUX4 target gene dysregulation, can be seen as early as in FSHD-affected fetuses, suggesting that FSHD pathology might occur early in muscle development [9]. Another explanation could be the particular vulnerability to DUX4 of the developing muscles and nervous system during childhood and puberty [10].

FSHD1 is the most common type (95%) of FSHD and accounts for almost all early-onset cases [11]. FSHD1 is a dominantly inherited disease caused by a genetic defect located in the chromosome 4q35 region, where variable deletions occur on a 3.3 kb tandem repeat (D4Z4) array, which can be visualized by *Eco*RI/B*ln*1 digested DNA fragment on the Southern blotting gel. In general, healthy individuals have 11 to more than 100 copies (*Eco*RI 50–300 kb) of D4Z4 repeats, whereas patients with FSHD1 have only ten or fewer D4Z4 repeats (<39 kb) on each copy of 4q35 [5,8,12]. Disruption of CpG methylation (hypomethylation), related to contraction of D4Z4 repeats, leads to chromatin relaxation of the 4q35 region, which initiates DUX4 production (Figure 1). Subsequently, the DUX4 expressed from the permissive chromosomal allele with a polyadenylation signal (PAS) results in transcriptional dysregulation and consequent degeneration of muscle cells [2,5]. FSHD2, representing less than 5% of patients with clinically defined FSHD, is not associated with deletions of D4Z4 copies but mutations of the *SMCHD1* (on chromosome 18p11) or *DNMT3B* gene (on chromosome 20q11) in the presence of a disease-permissive 4qA haplotype [6,8,13]. However, the comprehensive molecular genetic testing of FSHD2 is relatively complex and has not been made commercially available so far. Although there are different underlying genetic mechanisms between the two types of FSHD, they cannot be distinguished from clinical features.

In the majority of FSHD1 patients, the *Eco*RI fragment ranges from 15 to 38 kb. In contrast, patients with large contractions of D4Z4 (one to three D4Z4 repeats with *Eco*RI size of 10–14 kb) usually delineate the most severe phenotype and high frequency of systemic involvements [6,14,15,16]. On the other hand, as many as 30% of FSHD1 cases confirmed by genetic testing may be asymptomatic, especially those with small deletions (8–10 D4Z4 repeats), which delineate the mildest end of the disease severity spectrum [17]. Therefore, it is implied that the fragment size of D4Z4 is roughly negatively correlated with the clinical severity of FSHD [3,8,13,18,19].

## 3. Clinical Manifestations of FSHD

The onset of FSHD symptoms ranges from birth to 70 years old, and the clinical severity may vary from asymptomatic carriers to patients with extensive muscle wasting, leading to functional dependence and chronic respiratory failure [20]. FSHD is traditionally delineated as muscular dystrophy with slow progression, and the age of onset is variable with a 95% penetrance [3,4]. It usually starts with weakness in the facial, periscapular, and humeral muscle groups, except for the extraocular and deltoid muscles. Muscle weakness usually progresses in a descending manner, gradually involving the trunk, hip girdle, and muscle of distal lower extremities (ankle dorsiflexors). At the advanced stage of FSHD, about 20% (10–30%) of patients become wheelchair-dependent after the age of 50 [4,17]. However, around 20–30% of the genetically confirmed cases are asymptomatic, including both FSHD1 and FSHD2 patients [5,21]. These imply a more complex interplay of genetic and epigenetic factors in the manifestation of FSHD phenotypes.

Over time, affected individuals will develop a constellation of signs associated with FSHD, some of which strongly suggest FSHD, whereas other signs might more usually accompany other neuromuscular dystrophies and orthopedic anomalies. Besides, the involvement of other systems has been repeatedly reported in FSHD. Figure 2 illustrates an overview of FSHD-related signs and symptoms, including muscular and extramuscular involvement, in the different body regions. Facial involvement, usually representing the first clinical manifestation, is characterized by eye closure weakness, reduced facial expressions, transverse smile, difficulty in whistling, use of straws, or puckering the lips. The *i*nvolvement of the shoulder-girdle and humeral muscles can produce the features of scapular winging (high-riding scapula), horizontal clavicles, and limitation in shoulder abduction and elbow flexion. As the disease progresses or onset at a relatively early age, patients with a severely advanced phenotype may exhibit rare features, such as a poly-hill sign, Beevor’s sign, excessive curvature of the lumbar spine (hyperlordosis), Trendelenburg gait as well as drop foot [11,17]. Nevertheless, given that the bulbar, respiratory, and cardiac muscles are relatively unaffected by the disease course, patients with a typical FSHD type usually have a normal life expectancy [3,4,12].

Although FSHD is a hereditary neuromuscular disease, a negative family history certainly does not exclude its diagnosis. A significant percentage of cases, maybe up to 30%, are sporadic de novo mutations, and FSHD families frequently include asymptomatic gene carriers [4,20]. Besides, many patients do not fit this well-known classical FSHD phenotype. Atypical features have been reported, including less facial involvement, neck weakness, calf hypertrophy, and preferential involvement of lower limbs [22,23]. In particular, early-onset and late-onset cases are not uncommon, and the severity and sequence of involvement of different muscle groups may vary. Patients with FSHD may often exhibit non-specific discomforts, such as chronic musculoskeletal pain and fatigue [3,24]. Therefore, if healthcare providers do not actively ask and investigate symptoms and signs, it is easy to miss the diagnosis of FSHD.

## 4. Early-Onset Infantile FSHD

In general, there are two significant risk factors associated with disease severity of FSHD: (1) D4Z4 repeat size, as smaller D4Z4 repeat size is probably associated with a more rapid progression of disease course, and (2) age at onset, as the younger age at onset appears to be associated with a premature loss of ambulation [4,10,19]. Clinically, FSHD1 has been divided into classic FSHD, with a typical onset in the second decade (usually 15–30 years of age), and a severe early-onset (less than 10 years of age) called infantile FSHD. In general, infantile FSHD is regarded as a more severe, rapidly progressing variant of FSHD, which depends on genetic factors, mostly on the number of D4Z4 repeat loss.

Historically, the term “infantile FSHD” was initially proposed and considered a distinct subtype of FSHD based on clinical onset before two years of age [25]. The diagnostic criteria for infantile FSHD were further defined by affected patients with clinical onset of facial involvement before 5 years of age, plus manifestations of scapulohumeral weakness before 10 years of age [26]. Infantile FSHD accounts for about 10% (3 to 21%) of the total FSHD population with a general prevalence of one in 200,000 [20,26,27,28] and nearly 50% of the pediatric FSHD population [29]. Compared with the classic form, infantile FSHD almost invariably represents the severe end of the disease spectrum, with rapid muscle wasting and a higher prevalence of systemic involvement [16,20,30]. They often become wheelchair-dependent by the end of their first decade.

With the increase in the number of identified cases and the advent of genetic diagnosis, recent studies have concluded that the infantile FSHD is not a genetically separate entity but is part of the FSHD spectrum [16,17]. The link between the substantial deletion of D4Z4 repeat and infantile FSHD has been suggested. Very short *Eco*RI fragments (≤14 kb) are found in most infantile FSHD patients, but not all [19,20,30,31]. Furthermore, an association between a short *Eco*Rl fragment and systemic involvements, such as auditory and ophthalmological impairments, has been reported [16,20]. However, the correlation between clinical severity and D4Z4 fragment size is not consistent, and other genetic and environmental modifiers have been proposed [15,31]. Interestingly, the inheritance of infantile FSHD is often sporadic, and de novo mutations were more frequent in patients with infantile FSHD than classical FSHD [10,20]. Although positive family history is rare, patients with infantile FSHD obtaining a de novo mutation usually have unaffected carrier parents who carry somatic mosaicism of the mutation [12,13]. FSHD with somatic mosaicism of D4Z4 array lengths is more penetrant in males than in females [32]. Mosaic-affected males can already have a mild FSHD phenotype when only a small proportion of their cells carry an FSHD-sized allele, whereas females with a comparable ratio of affected cells and similar repeat sizes appear to be unaffected [33]. Significant clinical heterogeneity can also be observed in patients with infantile FSHD. There have been rare case reports describing children with infantile FSHD who had only mild limb involvement and maintain walking ability despite severe facial weakness [30,34]. Thus the clinical diagnosis of infantile FSHD does not always predict a more disabling outcome [15].

## 5. Peculiar Pattern of Muscle Involvement in Infantile FSHD

A recent systemic review indicated that patients with infantile FSHD were diagnosed with an average age of onset of 2.8 years, and 25% of them developed FSHD-related features in the first year of life [20]. Like the classical form, the first manifestation of infantile FSHD that alerts clinical attention is facial weakness, which may occur as early as infancy, and is often mistaken for Mobius syndrome, a congenital cranial dysinnervation disorder. A cross-sectional study of 52 FSHD1 patients who met the criteria for infantile FSHD showed that the onset time of facial weakness is positively correlated to the clinical severity [21]. In contrast, the age at the onset of shoulder weakness was not related to any functional motor index. The author suggested accurately capturing the onset age of facial weakness, for example, by reviewing family photos to reduce recall bias.

Asymmetric muscle involvement is commonly seen (about 50%) in classical and infantile FSHD [16,22]. However, the infantile FSHD patient population has a smaller variability in the severity of muscle wasting. About 40% of patients with infantile FSHD become wheelchair-dependent by the average age of 17 years [20], compared with 10% of the general FSHD population [4,17,35]. Moreover, the premature loss of ambulation in infantile FSHD patients also suggests a high prevalence of spinal deformities, especially lumbar hyperlordosis, caused by severely axial muscle weakness [20].

Recently, there have been emerging reports of the involvement of the lower limbs in the classic form of FSHD, especially in the hamstring and posterior calf muscles [36,37,38]. It is worth noting that in the anterior thigh muscle group (quadriceps), the rectus femoris seems to be more affected, especially in patients with early-onset FSHD [22,36]. We recently observed a peculiar thigh muscle involvement pattern in some patients with infantile FSHD who became wheelchair-bounded before 20 years of age. In the computed tomography (CT) image of their thighs, the selective degeneration of bilateral rectus femoris with relatively sparing other parts of anterior thigh muscles, accompanying dramatic wasting of hamstring muscles (Figure 3). Compared with other classical FSHD patients in our database, this special muscle imaging feature has never appeared at such a young age. As infantile cases are known to have a much faster disease progression, this unique pattern of muscular involvement may represent a “time-lapse picture” for classical FSHD with a slower disease course [10,16]. We suggest that this peculiar imaging finding of lower extremities in an FSHD patient may indicate a rapid motor function decline.

## 6. Respiratory Involvement in Infantile FSHD

A higher prevalence (8–11%) of chronic respiratory failure requiring assisted ventilation has been documented in the infantile FSHD patients [15,16,19,20,39], compared to the general FSHD populations (0–7%) [40,41,42]. A common finding of the restrictive ventilatory defect in patients with infantile FSHD implies an associated with extensive involvement of respiratory muscles, long-term wheelchair dependence, and spinal deformities (kyphoscoliosis) [15,16,20,39]. A correlation between the degree of compromised lung function (e.g., forced vital capacity) and early-onset FSHD has been recently identified [42]. Thus, regular pulmonary monitoring by spirometry in infantile FSHD patients is recommended [4,42]. However, a spirometry result must be interpreted cautiously in FSHD patients, as the poor sealing of the mouthpiece due to weakness of the perioral muscles (orbicularis oris) may cause biased air leakage [29]. In this case, a mask connected to the mouthpiece of spirometry may be applied to avoid air leaks [39].

## 7. Systemic Involvement with Extramuscular Features in Infantile FSHD

Approximately 50% of early-onset FSHD patients have systemic features (Figure 2). Of these, about two-thirds of infantile FSHD patients had a single extramuscular feature, and the other third had multiple systemic features [20]. A genotype–phenotype correlation between the contraction of D4Z4 and the frequency of systemic involvement has been proposed, especially in neurosensory hearing loss and brain dysfunction (intellectual disability and epilepsy) [16,19,20,43]. These two features are also the most reported first extramuscular features in children with infantile FSHD [16,19]. However, an updated study refutes this correlation, indicating that the disease severity may not be explained by the D4Z4 array size alone [10]. A potential interaction between D4Z4 repeat size and other modifying factors may involve the pathomechanism [44]. Another explanation could be the particular vulnerability to DUX4 of the developing muscles and nervous system during childhood and puberty, which argues for the developmental instead of degenerative pathophysiology of infantile FSHD.

### 7.1. Auditory Impairment

The prevalence of hearing loss in the general FSHD population is unclear, but it may be related to the size of D4Z4 repeat (especially in those with EcoRI/BlnI ≤ 20 kb) [43]. Indeed, the most common extramuscular involvement in infantile FSHD is sensorineural hearing loss, which accounts for 40% of patients, especially high-frequency loss [19,20]. Hearing aids are rarely needed, except for those with early onset of FSHD and large D4Z4 contractions [4,16]. Of note, about 7% of infantile FSHD patients may have subclinical hearing loss at auditory testing [20]. Due to undetected hearing loss that may jeopardize childhood language development, the current guideline has recommended that children with infantile FSHD should have audiometry examination at diagnosis and annually after that until entry into the school [4]. If the speech development of older children with FSHD is normal, no formal assessment is required, and hearing monitoring is carried out routinely in school [45].

### 7.2. Retina Vasculopathy

Retinopathy, mainly tortuous retinal vessels, has been reported in about 37% of patients with infantile FSHD [17], lower than that (50–75%) in the general FSHD population [46]. Rarely, optic nerve atrophy may present [16]. However, the exact prevalence of retinal vascular abnormalities in patients with infantile FSHD may be underestimated because retinal fluorescence angiography is usually not arranged for pediatric cases [14,16,29]. Another explanation might be the later onset of retinal abnormalities [20].

Although retinal vascular tortuosity in FSHD patients is mostly subclinical, in infantile cases with large D4Z4 contractions, it may progress to Coats disease, a severe retinal vasculopathy complicated with exudative leakage that can cause retinal detachment and blindness [47]. Indeed, the prevalence of clinical vision loss is higher in infantile FSHD patients than in the classical FSHD ones, 6% and 0.8–1.7%, respectively [17]. As Coats disease is potentially preventable by early laser photocoagulation, it is recommended that infantile FSHD patients should be screened through dilated indirect ophthalmoscopy, and fluorescence angiography should be further arranged if any symptoms of visual deterioration are reported [4,16,29]. Otherwise, children with unexplained retinal vasculopathy should undergo a thorough neuromuscular assessment to rule out the diagnosis of infantile FSHD [17].

### 7.3. Brain Dysfunction

Central nervous system involvement (epilepsy and intellectual disability) is a rare extramuscular manifestation in the general FSHD population and seems to be only associated with early-onset FSHD [14,48,49]. Epilepsy was present in 8% of infantile FSHD patients and was always accompanied by intellectual disability [14,16,20]. Rarely, epilepsy can initially present with infantile spasms [16,50,51]. Once the infantile FSHD is diagnosed, education about seizure recognition and safety during epilepsy attacks should be conveyed; for patients with suspected seizures, electroencephalogram (EEG) should be indicated [17]. However, the presence of severe cognitive impairment may mask the onset of facial weakness and hearing or visual impairment in early childhood [52]. Given that up to 15% of infantile FSHD patients are known to have developmental delay [19,20], it is recommended that visual and auditory assessments and timely interventions be taken to ensure that these children obtain the optimal developmental outcomes [17].

### 7.4. Cardiac Involvement

Abnormal electrocardiography (ECG) is reported in about 9% of infantile FSHD patients. This prevalence roughly corresponds to the general FSHD population, including findings of arrhythmia and other minor abnormalities [20]. Like classical FSHD, minor conductive abnormalities, particularly incomplete right-bundle branch block (RBBB), are the most documented cardiac involvements in the infantile FSHD patient group [53,54]. The clinical consequences of arrhythmia appear to be minor, as there are no data showing patients requiring pacemaker intervention. Notably, these ECG abnormalities were mainly found in patients of Asian origin, suggesting geographic or ethnic differences [20].

However, there is a possibility of incomplete RBBB progressing into a fatal complete form, which has also reported an association in 60% of patients with classical FSHD who manifested cardiac symptoms [53]. Given that children may not express their cardiac symptoms, an initial baseline study and regular follow-up assessment of echocardiogram and ECG should be recommended for those diagnosed with infantile FSHD. Otherwise, there is no need for routine cardiac screening in the absence of any cardiac symptoms in the general FSHD population [17].

## 8. Laboratory Findings and Potential Biomarkers of FSHD

Serum creatine kinase (CK) in symptomatic FSHD patients may rise to five times the upper limit of normal. The electromyogram usually shows a diffuse myopathic feature with sporadic fibrillations and positive sharp waves. Motor and sensory nerve conduction studies are usually unremarkable. Muscle biopsy may show inflammatory cellular infiltrates in approximately 40% of FSHD patients and may be extensive in early-onset infantile FSHD [3].

Although definitive diagnosis of FSHD is made with molecular genetic testing, alternative biomarkers—available in less invasive procedures, such as blood collection (rather than muscle biopsy)—will be reliable outcome measures and disease surveillance, especially in children with a severe infantile FSHD variant. Several biomarkers have been proposed, focused on DUX4 target gene expression and genes found to be differentially expressed in FSHD muscle biopsies. Several DUX4 regulated genes were aberrantly expressed in FSHD muscle but not in the control muscle [55]. As many of these DUX4 regulated genes are not normally expressed in skeletal muscle, they are ideal candidate biomarkers for FSHD [56]. Several studies have demonstrated that DUX4 is a significant biomarker of FSHD status and is associated with disease severity using the expression of derivatives of a single patent-pending DUX4 target gene biomarker [57,58]. However, an updated study shows that PAX7 target gene expression is a superior FSHD biomarker than the DUX4 target gene approach, associating with pathological severity. Besides, proteomics findings and the miRNA of patient sera are potential biomarker candidates to predict the disease state and correlate with disease severity in FSHD [59,60]. Nevertheless, long-term studies that monitor their levels beginning from a young age are still lacking, representing an authentic biomarker for infantile FSHD. Further confirmatory studies are still ongoing.

## 9. Emerging Therapeutic Approaches for FSHD

To date, there are no pharmacological disease-modifying treatments or interventions known to halt the progression of FSHD or reverse muscle dystrophies in patients. However, recent advances in the discovery of the complex molecular mechanisms of FSHD have led to a better understanding of the pathomechanism and allowed the development of targeted therapies, especially in the epigenetic and transcriptional regulation of the aberrantly expressed DUX4 in FSHD [12]. Although most classical FSHD patients have a normal life expectancy, the early-onset infantile variant may have higher morbidities and mortalities, not only because of a rapid progression of muscle wasting but a high rate of bulbar, respiratory, and cardiac involvement. Therefore, patients with infantile FSHD may need a novel disease-modifying therapy more urgently than we expected. Table 2 summarizes the current ongoing preclinical and clinical trials of potential treatments for FSHD. In general, emerging therapeutic trials are either directed at increasing muscle strength or slowing disease progression. Nevertheless, a reliable disease pathomechanism model is still needed to develop targeted treatments [2].

## 10. Standard Care for Patients with Infantile FSHD

To date, management in patients with FSHD is mainly supportive. While minimally symptomatic adults may require infrequent follow-up and minimal intervention, patients with infantile FSHD require closer monitoring by a multidisciplinary care team [4,74], with input from various specialties (Figure 4). The mainstay of a standard of care includes physical therapy, pain control, orthopedic interventions, and surveillance and interventions targeting FSHD-related extramuscular complications, which involve different specialists, including an ophthalmologist, otologist, speech therapist, cardiologist, and pulmonologist. Once a child with infantile FSHD reaches skeletal maturity, the operation for scapular fixation performed by an experienced multidisciplinary care team is potentially safe and effective for carefully selected patients [17]. Other anticipatory care includes screening for extramuscular manifestations, particularly among children with infantile FSHD.

## 11. Conclusions

Patients with infantile FSHD have a higher risk of rapid decline in motor function, the early loss of independent ambulation, as well as systemic involvement. In the past decade, there have been increasing advances in the comprehension of the pathophysiology of FSHD. It is hoped that these advances may help identify possible therapeutic targets in the coming years. Nevertheless, early diagnosis, preserving support following the standards of care guidelines, and cooperation with global FSHD registries are essential to optimize the outcome of affected individuals and prepare for clinical trials. Additionally, efforts are being made to establish a patient registry and validated biomarkers and clinical outcome measures to improve patient enrollment into future clinical trials and reliably monitor treatment response.

## Figures and Tables

**Figure 1 ijms-21-07783-f001:**
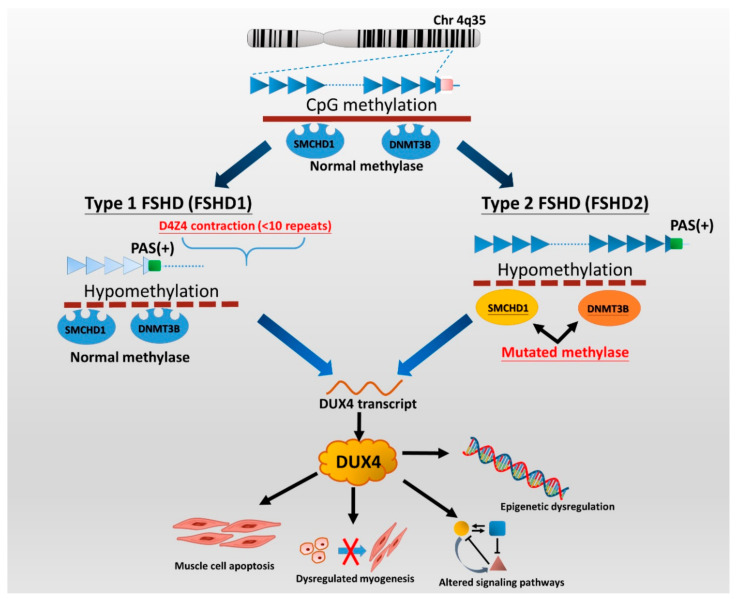
The genetic pathomechanisms of FSHD types 1 and 2. Normal individuals have 11 to more than 100 repeats (50–300 kb) of D4Z4 repeats on the sub-telomeric region of chromosome 4q35, whereas individuals with FSHD1 have a deletion of the array to 10 or fewer D4Z4 units (<39 kb). Contraction of D4Z4 repeats with the presence of a permissive 4qA haplotype or allele containing a polyadenylation signal (PAS, green box) then disrupts CpG methylation (hypomethylation), leading to chromatin relaxation at the array and allowing for the expression of a toxic transcription factor, DUX4 (double homeobox 4). Additionally, the DUX4 retrogene within the D4Z4 unit lacks a stabilizing PAS but can use an additional exon immediately distal to the repeat that contains a polyadenylation signal (DUX4-PAS). In FSHD2, mutation of methylase-related genes, *SMCHD1* or *DNMT3B,* also causes hypomethylation of D4Z4 chromatin structure, resulting in DUX4 expression. There are several potential mechanisms of DUX4 involving in FSHD pathogenies. DUX4 has been implicated as being involved in apoptosis, myogenesis, epigenetic regulation, and regulatory signaling pathways in skeletal muscle and extramuscular tissues, including retinal vessels (Wnt/β-catenin pathway) and auditory cells (JNK pathway) [5,6,11]. Picture modified from Lek et al. 2015 [2] and Lim et al. 2020 [5].

**Figure 2 ijms-21-07783-f002:**
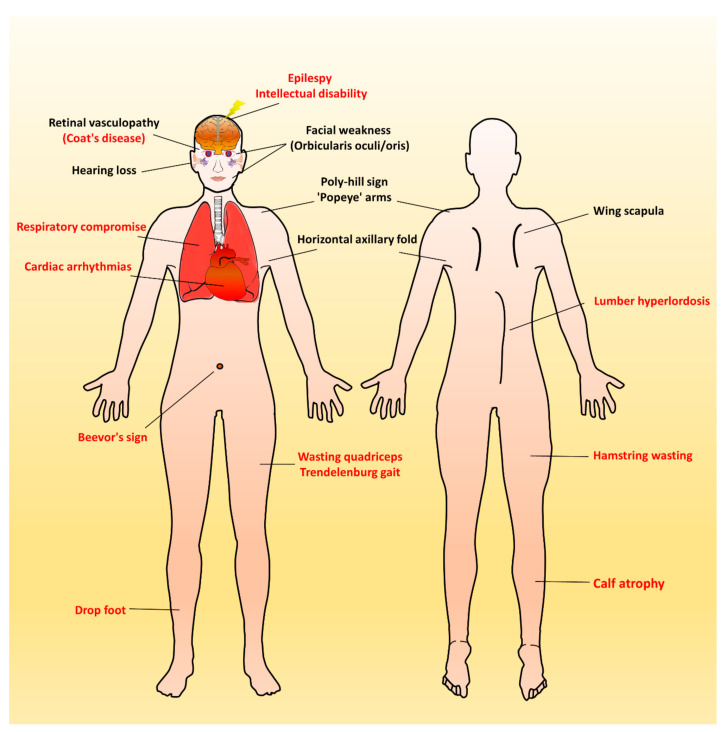
Characteristic muscular and extramuscular signs of FSHD. Generally, most patients with FSHD have multiple distinctive features (as listed in Table 1). Illustrated FSHD-related signs and symptoms marked with red text indicate atypical and relatively rare features that are usually presented in infantile FSHD or at the advanced stage of disease course. Figure modified from Mul et al. 2016 [11].

**Figure 3 ijms-21-07783-f003:**
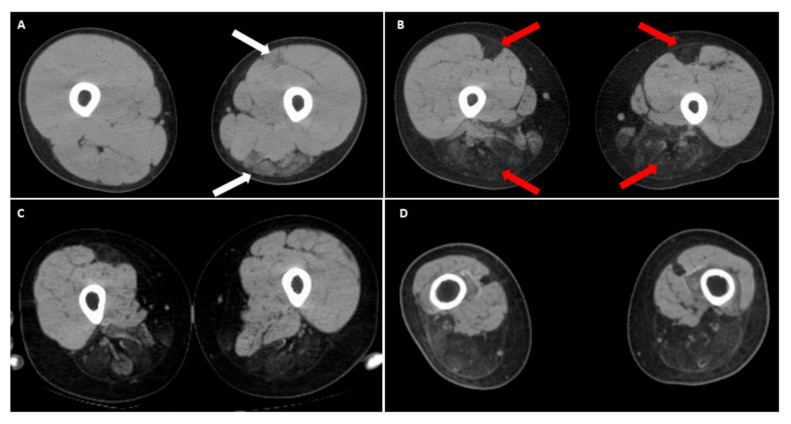
Muscle computed tomography (CT) of upper thighs in infantile FSHD patients. (**A**) A male with 13 kb E*co*RI showing early degeneration of left hamstring muscles and rectus femoris (white arrows) at 12 years old, compared to a relatively normal appearance of right thigh muscles; (**B**) Follow-up at 15 years old showing a rapid progression of muscle degeneration depicting a “wrench-head” appearance caused by extensive muscle wasting at the bilateral hamstring muscles with selectively affected rectus femoris of quadriceps (red arrows). Another two CT findings of (**C**) right leg of a 16-year-old female (11 kb *Eco*RI), and (**D**) both legs of a 28-year-old male (10 kb *Eco*RI), showing a similar muscle pattern, respectively. Informed consents were obtained from above reported patients.

**Figure 4 ijms-21-07783-f004:**
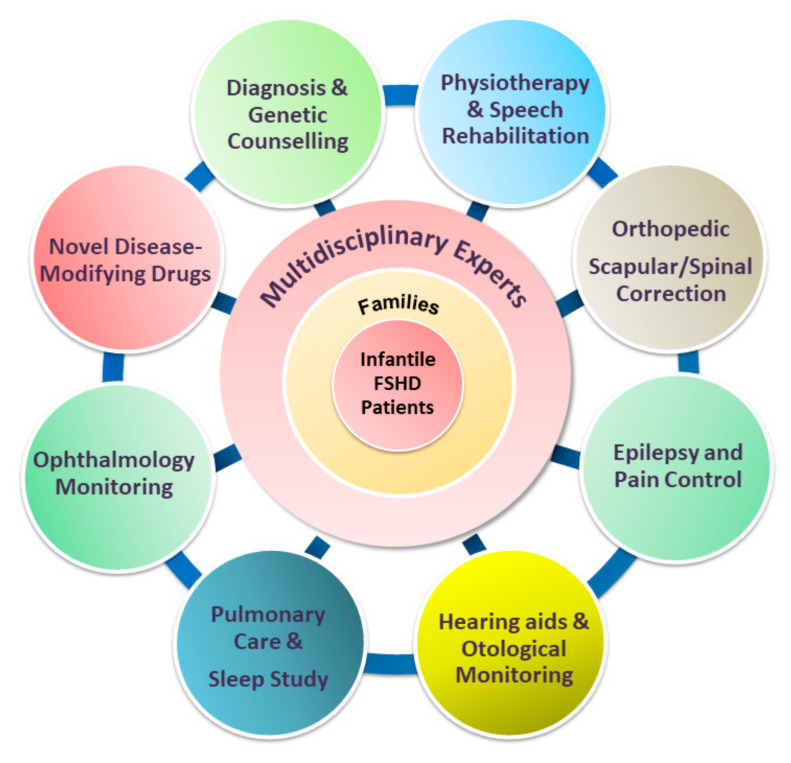
The paradigm of multidisciplinary care of infantile FSHD, incorporating disease-modifying therapies with the standard of care (SOC). Novel disease-modifying agents and evolving multidisciplinary supportive management need to occur concomitantly to achieve the best possible outcome for patients with infantile FSHD. Implementation of comprehensive SOC also plays a critical role in drug development because it may eliminate the variability of treatment outcomes due to variable care received.

**Table 1 ijms-21-07783-t001:** Diagnostic criteria for facioscapulohumeral dystrophy (FSHD).

**Inclusion Criteria**	Facial, humeral, and shoulder girdle muscle weakness (in familial cases, facial weakness is present in >90% of affected individuals) A weakness of shoulder-girdle muscle groups more prominent than hip-girdle ones (not applicable in severely early-onset cases) Autosomal dominant inheritance in familial cases
**Exclusion Criteria**	Extraocular or pharyngeal muscle weakness Prominent and diffuse elbow contractures Cardiomyopathy Distal symmetrical sensory loss Dermatomyositic rash or signs of an alternative diagnosis Electromyographic evidence of myotonia or neurogenic potentials
**Supplementary Criteria**	Asymmetry of muscle weakness A descending sequence of involvement Early, often partial, abdominal muscle weakness (positive Beevor’s sign) Sparing of deltoid muscles Typical shoulder profile: straight clavicles, forward sloping of shoulders Relative sparing of neck flexors A selective weakness of wrist extensors in distal upper extremities Sparing of calf muscles High-frequency hearing loss Retinal vasculopathy

Source: Facioscapulohumeral Consortium at the International Conference on the Cause and Treatment of Facioscapulohumeral Dystrophy, Boston, 1997. Data presentation is modified from Liew et al. 2015 [3].

**Table 2 ijms-21-07783-t002:** Therapeutic approaches in facioscapulohumeral dystrophy: current clinical trials and preclinical strategies.

Therapeutic Approach (Mechanism of Action)	Compound Agents	Experiment Description/Potential Effectiveness	Trial Phase	Reference
Knocking down the aberrantly expressed DUX4	Antisense oligonucleotides (AON) against DUX4	Experiments showing morpholinos can either reduce DUX4 expression or inhibit translation of a DUX4-regulated gene, paired-like homeodomain transcription factor 1 (Pitx1)	Preclinical	[61,62]
	miRNA against DUX4	Adeno-associated viruses (AAV)-vector miRNA against DUX4 into mouse muscles ectopically expressing DUX4 and was able to reduce DUX4 and improve pathologies induced by it	Preclinical	[63]
Reduce DUX4 expression by suppressing p38, mitogen-activated protein kinases (MAPKs) pathway	Losmapimod (p38α/β inhibitor)	In a preclinical study, p38 inhibitors effectively suppress DUX4 expression in a mouse xenograft model of human FSHD gene Regulation In a phase 1 study, losmapimod was well-tolerated and achieved dose-dependent exposure in plasma and muscle at concentrations to provide efficacy by reducing DUX4 activity. These results support advancing the dose of 15 mg twice daily into phase 2.	Phase 1/2 (NCT04003974, NCT04004000, and NCT04264442), active	[64,65]
Immune modulation to reduce FSHD-related inflammatory response in muscles	ATYR1940	ATYR1940 is a physiocrine-based protein (histidyl tRNA synthetase) shown to modulate immune responses in skeletal muscles Improvement of individualized neuromuscular quality of life (INQoL) after 12-weeks treatment of intravenous 3 mg/kg ATYR1940 in adult FSHD patients Improvement of manual muscle testing (MMT) after 12-weeks treatment in adolescents and young adults with early-onset FSHD	Phase 1b/2a (NCT02836418), completed	[66,67]
Activating compensatory pathways (Myostatin inhibitors)	MYO-029	Neutralizing antibody against myostatin Clinical study: showing good safety and tolerability, but no significant improvement of muscle strength or function	Phase 1/2 (NCT00104078), completed	[68]
	ACE-083	Inhibitor of activins and myostatin, shown to facilitate muscle growth Phase 1 trial showing well-tolerated and associated with dose-dependent increases in muscle volume Phase 2 trial was terminated in 2019 as ACE-083 did not achieve functional secondary endpoints in the trial.	Phase 1 (NCT02257489), completed Phase 2 (NCT02927080): terminated	[69]
DUX4-related oxidative stress	Antioxidants (vitamin C, vitamin E, zinc gluconate, and selenomethionine)	Oxidative stress has been proposed as a downstream effect of DUX4 Phase 3 study of a combination of antioxidants showed some benefit of maximal voluntary contraction and endurance limit time, but no improvement in 2-minute-walk test	Phase 3 (NCT01596803), completed	[70]
Target other molecular pathways related to DUX4	Tyrosine kinase inhibitor (sunitinib) Poly (ADP-ribose) polymerase (PARP1) inhibitors 17-estradiol (E2) β2-adrenergic agonists	Sunitinib can inhibit RET signaling and rescue differentiation in both mouse myoblasts expressing DUX4 and FSHD patient-derived myoblasts 17-estradiol (E2) shows to improve the differentiation of FSHD myoblasts β2 agonists considerably inhibited DUX4 expression	All are preclinical.	[71,72,73]

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
