# Peer review of "Early-Onset Infantile Facioscapulohumeral Muscular Dystrophy: A Timely Review"

_ijms, 2020, doi:10.3390/ijms21207783_

Round 1

Reviewer 1 Report

Review of the INJS article_Infantile FSHD_2020

The article gives an excellent overview of the clinical symptoms of FSHD including a short genetic background. It focuses on the heterogenous infantile form of the disease with some peculiar clinical features.

  1. Lines 51-69. Genetic background, e.g. epigenetic modulation of the DUX4 retrogene and its impact on neighbouring genes, the recent knowledge and models on pathomechanism should be explained in more details as it has strong association with the aim of the review. In general, epigenetic and transcriptional regulation of the aberrantly expressed DUX4 in FSHD should be discussed, especially in possible association of the classic and infantile FSHD phenotype.
  2. Figure 1, lines 87-88: “permissive 4qA haplotype, called polyadenylation signal (PAS, green box)”. It is more appropriate to use “permissive 4qA haplotype or allele containing a polyadenylation signal (DUX4 PAS)”. This might need some detailed explanation e.g. that DUX4 retrogene within the D4Z4 unit lacks a stabilizing polyadenylation signal (PAS) but can make use of an additional exon immediately distal to the repeat that contains a stabilizing DUX4-PAS.
  3. Lines 91-95, Figure 1. Reference for the potential mechanisms of DUX4 in pathogenesis of FSHD should be given.
  4. Lines 101-102. Some explanation is needed here that the penetrance is 95% by the age of 20 but around 20% of the genetically confirmed cases are asymptomatic. See also line 123 where this phenomenon is included.
  5. Lines 114-115. “The involvement of the shoulder-girdle and humeral muscles can produce the features of the high-riding scapula” should not be written in
  6. Lines 144-145. “However, it is still unclear whether infantile FSHD is inherently a more severe, rapidly progressing disease or if these patients are more severe as a result of longer disease duration” - the second part of the sentence does not really make sense as it is a consequence of the earlier onset, which depends on genetic factors, mostly on the number of D4Z4 repeat loss. This is later discussed properly in the article. Additionally, patients cannot be more severe, only their phenotype.
  7. Lines 177-178. The sentence needs rewording “….data showed that the earlier the onset of facial weakness was associated with the greater disease severity.”
  8. Lines 155-170. Somatic mosaicism should be explained somewhat more extensively. It should also be mentioned that FSHD with somatic mosaicism of D4Z4 array lengths is more penetrant in males than in females and possible reasons could be discussed.
  9. Lines 319-321. Some lines have formatting errors.

Author Response

1. Lines 51-69. Genetic background, e.g. epigenetic modulation of the DUX4 retrogene and its impact on neighbouring genes, the recent knowledge and models on pathomechanism should be explained in more details as it has strong association with the aim of the review. In general, epigenetic and transcriptional regulation of the aberrantly expressed DUX4 in FSHD should be discussed, especially in possible association of the classic and infantile FSHD phenotype.
Reply: We appreciated Reviewer 1’s comment. As for your suggestion, we have added the description regarding the role of epigenetic and transcriptional regulation of DUX4 in FSHD pathogenesis in the revised paragraph (revised manuscript pages 6 & 7). Moreover, we have also added the relevant reference referring readers to get more knowledge about the genetic background of FSHD and its infantile variant.

2. Figure 1, lines 87-88: “permissive 4qA haplotype, called polyadenylation signal (PAS, green box)”. It is more appropriate to use “permissive 4qA haplotype or allele containing a polyadenylation signal (DUX4 PAS)”. This might need some detailed explanation e.g. that DUX4 retrogene within the D4Z4 unit lacks a stabilizing polyadenylation signal (PAS) but can make use of an additional exon immediately distal to the repeat that contains a stabilizing DUX4-PAS.
Reply: As for your suggestion, we have added this discussion in the Figure 1 legend (in red text).

3. Lines 91-95, Figure 1. Reference for the potential mechanisms of DUX4 in pathogenesis of FSHD should be given.
Reply: We have added the references.

4. Lines 101-102. Some explanation is needed here that the penetrance is 95% by the age of 20 but around 20% of the genetically confirmed cases are asymptomatic. See also line 123 where this phenomenon is included.
Reply: We have added an explanation about the asymptomatic cases with confirmed genetic diagnosis at the end of the paragraph (Page 10, lines 9-12).

5. Lines 114-115. “The involvement of the shoulder-girdle and humeral muscles can produce the features of the high-riding scapula” should not be written in.
Reply: We have corrected this description.

6. Lines 144-145. “However, it is still unclear whether infantile FSHD is inherently a more severe, rapidly progressing disease or if these patients are more severe as a result of longer disease duration” - the second part of the sentence does not really make sense as it is a consequence of the earlier onset, which depends on genetic factors, mostly on the number of D4Z4 repeat loss. This is later discussed properly in the article. Additionally, patients cannot be more severe, only their phenotype.
Reply: We have modified the description in the sentence (Page 13).

7. Lines 177-178. The sentence needs rewording “….data showed that the earlier the onset of facial weakness was associated with the greater disease severity.”
Reply: We have amended the description of this sentence (Page 15).

8. Lines 155-170. Somatic mosaicism should be explained somewhat more extensively. It should also be mentioned that FSHD with somatic mosaicism of D4Z4 array lengths is more penetrant in males than in females and possible reasons could be discussed.
Reply: We have added more discussion and references regarding the mosaicism of the D4Z4 array and its association with gender (Page 14, lines 10-14).

9. Lines 319-321. Some lines have formatting errors.
Reply: We have corrected it.

Reviewer 2 Report

The authors have put together a comprehensive literature review of known facts on infantile FSHD. While there are many reviews on adult FSHD, there does not seem to be many that specifically focus on infantile FSHD and thus this review is needed in the field. The figures are appropriate and informative for the topics discussed. Minor changes suggested below:

For Figure 3 - this would benefit from a 'control' CT picture from a healthy/unaffected individual for comparison.

For Table 2: Please add current clinical trial details by Fulcrum on their drug Losmapimod (MAPK inhibitor). Preclinical studies also in this paper: https://www.ncbi.nlm.nih.gov/pmc/articles/PMC6652132/

Additionally, the review would benefit from a short discussion on disease biomarkers in infantile FSHD if they have been looked at (i.e. DUX4 target genes). Are they the same set found in adult FSHD? How early do they appear and are they correlated with severity? Are there long-term studies that monitor their levels beginning from a young age?

There are also minor grammatical issues and awkward sentence wording in certain places of the text.

Author Response

1.For Figure 3 - this would benefit from a 'control' CT picture from a healthy/unaffected individual for comparison.
Reply: We appreciate Reviewer 2’s comment. Due to the study design and inform consent approved by IRB, we did not obtain the CT picture of normal individuals. However, as shown in Figure 3A presenting status quo before significant degeneration of right thigh, we think it could alternatively represent the “control” CT picture compared to the subsequently peculiar denegation pattern of muscle groups.

2.For Table 2: Please add current clinical trial details by Fulcrum on their drug Losmapimod (MAPK inhibitor). Preclinical studies also in this paper: https://www.ncbi.nlm.nih.gov/pmc/articles/PMC6652132/
Reply: We sincerely appreciate Reviewer 2 providing such important information about the novel therapeutic trial in FSHD. As for your suggestion, we have added this information and its relevant references in the revised Table 2.

3.Additionally, the review would benefit from a short discussion on disease biomarkers in infantile FSHD if they have been looked at (i.e. DUX4 target genes). Are they the same set found in adult FSHD? How early do they appear, and are they correlated with severity? Are there long-term studies that monitor their levels beginning from a young age?
Reply: In the revised manuscript (page 23), we have added a discussion about the DUX4 target gene as potential biomarkers in FSHD. We also describe an updated study regarding PAX7 target gene expression, which seems to be a superior FSHD biomarker than the DUX4 target gene approach, associating with pathological severity. However, there are still lacking long-term studies that monitor their levels beginning from a young age, representing an authentic biomarker for infantile FSHD.

4.There are also minor grammatical issues and awkward sentence wording in certain places of the text.
Reply: We have corrected these errors.
